# Features of Appendix and the Characteristics of Appendicitis Development in Children with COVID-19

**DOI:** 10.3390/biomedicines12020312

**Published:** 2024-01-29

**Authors:** Grigory Demyashkin, Konstantin Gorokhov, Vladimir Shchekin, Matvey Vadyukhin, Artem Matevosyan, Arina Rudavina, Anna Pilipchuk, Alina Pilipchuk, Svetlana Kochetkova, Dmitrii Atiakshin, Petr Shegay, Andrey Kaprin

**Affiliations:** 1Department of Pathomorphology, National Medical Research Centre of Radiology, Ministry of Health of Russia, 249036 Obninsk, Russia; auts77@gmail.com (K.G.); dr.shegai@mail.ru (P.S.); kaprin@mail.ru (A.K.); 2Laboratory of Histology and Immunohistochemistry, I.M. Sechenov First Moscow State Medical University (Sechenov University), 119048 Moscow, Russia; dr.dga@mail.ru (G.D.); vma20@mail.ru (M.V.); aretem252150@mail.ru (A.M.); rudavinaarina@yandex.ru (A.R.); alina.150164@list.ru (A.P.); alina.pilipchuk@list.ru (A.P.); sv.k0ch@yandex.ru (S.K.); 3Research and Educational Resource Center for Immunophenotyping, Digital Spatial Profiling and Ultrastructural Analysis Innovative Technologies, Peoples’ Friendship University of Russia (RUDN University), 117198 Moscow, Russia; atyakshin-da@rudn.ru; 4Department of Urology and Operative Nephrology, Peoples’ Friendship University of Russia (RUDN University), 117198 Moscow, Russia

**Keywords:** acute appendicitis, COVID-19, children, immunohistochemistry, CD

## Abstract

Background: Research on the subject of the influence of SARS-CoV-2 mechanisms on human homeostasis remains an actual problem. Particular interest is the study of pathomorphological changes in the appendix in children with COVID-19. Objectives: Aim of this study: morphological and molecular biological evaluation of the appendix in children of different age groups with COVID-19. Methods: Groups were formed on the basis of anamnestic, clinical, and morphological data: I (*n* = 42; aged 2 to 18 years, average age—10.8 ± 4.79)—with an established clinical diagnosis: coronavirus infection (COVID-19; PCR+); II (*n* = 55; aged 2 to 18 years, average age—9.7 ± 4.77)—with a confirmed clinical diagnosis of acute appendicitis; collected before the onset of the COVID-19 pandemic in 2017–2019; and III (*n* = 38; aged 2 to 18 years, average age—10.3 ± 4.62)—the control group. Histological and immunohistochemical studies were conducted using primary antibodies to CD3, CD4, CD68, CD163, CD20, and CD138 and to pro-inflammatory (IL-1, IL-6) and anti-inflammatory (IL-4, IL-10) cytokines. Results: In most samples of appendixes in children with COVID-19, signs of destructive phlegmonous–ulcerative and gangrenous appendicitis were discovered. An increase in CD3+, CD4+, CD68+, CD163+, and CD20+ CD138+ immunocompetent cells was found in the appendix of children with COVID-19. As well, there was an increase in pro-inflammatory (IL-1, IL-6) and anti-inflammatory (IL-4, IL-10) cytokines. Conclusions: The aforementioned pathological and immunohistochemical changes were more pronounced in the group of children aged 6–12 years (childhood).

## 1. Introduction

At the end of 2019, cases of pneumonia caused by a new coronavirus infection were registered in Wuhan. The rapid spread of SARS-CoV-2 in a short period of time led to over 46 million cases of infection and approximately 1.2 million deaths [1]. Patients can carry the infection asymptomatically but can also develop severe life-threatening pneumonia with multi-organ failure.

The entry gate for the virus is primarily the epithelium of the upper respiratory tract [2]. The virus enters the cell by binding to specific receptor proteins: angiotensin-converting enzyme 2 (ACE-2) and serine protease Furin [3].

At the same time, there is damage to the respiratory, cardiovascular, gastrointestinal, genitourinary, reproductive, and other systems, which determines a diverse clinical picture in patients [4,5].

The incidence of COVID-19 in pediatric patients is of particular interest. At the beginning of the COVID-19 pandemic, it was believed that children were less susceptible to SARS-CoV-2 infection. However, subsequent studies have shown that children and adolescents are susceptible to infection, and most have mild or asymptomatic courses [6,7].

Some authors report cases of severe course when children develop multisystem inflammatory syndrome (MIS-C). This syndrome mainly manifests as gastrointestinal organ damage, as it has been established that SARS-CoV-2 can invade intestinal epithelial cells. The clinical features of MIS-C are similar to many other inflammatory diseases observed in children, such as toxic shock syndrome, macrophage activation syndrome, and Kawasaki disease [8,9,10].

Now, there are reports in the literature considering children and adolescents infected with the new coronavirus, whose clinical picture can be determined as acute abdomen and appendicitis. In some studies, there were also individual cases of pseudoappendicitis and acute surgical abdomen in children with positive SARS-CoV-2 results [11,12].

Considering the lack of data on the possible invasion of SARS-CoV-2 in the epithelium of the appendix in children [13], as well as the obvious interest in the features of the course of acute appendicitis in these conditions, research in this direction seems relevant.

Therefore, a more detailed study of the effect of SARS-CoV-2 on the development of acute appendicitis in children is necessary. Currently, only isolated clinical cases have been described in the literature, but there are no detailed morphological and molecular biological studies correlating new coronavirus infection and acute appendicitis in children. In this study, we attempted to expand the understanding of the pathogenesis of acute appendicitis in children with COVID-19 by performing detailed morphological, immunohistochemical, PCR, and fluorescence in situ hybridization (FISH) research.

## 2. Research Objective

Morphological and molecular biological evaluation of the appendix in children of different age groups with COVID-19.

### Research Tasks

A comparative morphological analysis of the appendix in children of different age groups with COVID-19.Determine the cellular character of the inflammatory infiltrate in children of different age groups with COVID-19 based on the levels of expression of immunocompetent cell differentiation clusters: CD3, CD4, CD20, CD68, CD163, and CD138.Evaluate the state of the cytokine balance in the tissues of the appendixes of children of selected age groups with COVID-19 based on the levels of expression of pro-inflammatory (IL-1, IL-6) and anti-inflammatory (IL-4, IL-10) markers.

## 3. Materials and Methods

Study type: analytical, “case-control”.

Experimental design. According to the WHO age classification, as well as anamnestic, clinical (symptoms, nasopharyngeal swab, chest CT), and morphological data, the following groups were formed:(*n* = 42; aged 2 to 18 years, average age—10.8 ± 4.79)—surgical material of vermiform appendixes after appendectomy in children with a confirmed clinical diagnosis of coronavirus infection (COVID-19, PCR+).(*n* = 55; aged 2 to 18 years, average age—9.7 ± 4.77)—surgical material of vermiform appendixes after appendectomy in children with a confirmed clinical diagnosis of acute appendicitis; collected before the onset of the COVID-19 pandemic in 2017-2019.(*n* = 38; aged 2 to 18 years, average age—10.3 ± 4.62)—the control group. For the purity of this study, autopsy material of the vermiform appendix was obtained before the pandemic in 2017–2019. Death occurred due to decompensation or complications of a number of diseases. Macroscopic signs of inflammatory and/or oncogenic processes were absent.

This study was approved by the Ethical Committee of National Medical Research, Radiological Centre of the Ministry of Health of the Russian Federation (Protocol No. 3; 7 April 2022). All the actions complied with the Declaration of Helsinki (WMA Declaration of Helsinki–Ethical Principles for Medical Research Involving Human Subjects, 64th WMA General Assembly, Fortaleza, Brazil, October 2013) (Table 1).

The diagnosis in admission for all patients in Group I was a new coronavirus infection (confirmed by computer morphometry and nasopharyngeal swab PCR+); the CT degree varied depending on the severity and duration of the disease, as well as the age of the patients.

For inclusion in this study, a careful analysis of medical histories was conducted to exclude the possible influence of chronic diseases or medication use to obtain reliable results. Patients received only dexamethasone, heparin, and oxygen therapy.

Morphological study. Assessment of the appearance of the vermiform appendix and mesoappendix after extraction. Fragments were placed in a 10% formalin solution with phosphate buffer for 72 h. Dehydration was made using a series of ascending concentrations of alcohol in the tissue histoprocessing apparatus from “Pool Scientific Instruments” (Zurich, Switzerland), followed by embedding in paraffin. Paraffin blocks were then sectioned (at least 20 sections per block), and 3 μm thick sections were placed on regular glass slides or specialized adhesive polylysine slides (Super Frost Plus, “Mainzel Glaser”, Polylisine, Braunschweig, Germany). The sections were stained with Mayer’s hematoxylin and eosin and prepared for histochemical and immunohistochemical investigations on special adhesive slides.

Microscopic analysis was performed using a video microscopy system (Leica DM2000 microscope, Wetzlar, Germany; Leica ICC50 HD camera).

An immunohistochemical investigation was made after deparaffinization and rehydration of the paraffin sections using a standard protocol in the automated immunostainer Bond-Max (Leica, Wetzlar, Germany). Primary mouse monoclonal antibodies, Ready-to-Use (RTU, Leica, Wetzlar, Germany), against CD3 (LN10), CD4 (4B12), CD20 (MJ1), CD68 (514H12), CD163 (10D6), and CD138 (MI15), were used.

The number of CD+ cells was determined by computer morphometry in 10 fields of view with a total area of 1.6 mm^2^ at a magnification of 400×. The quantitative density of CD+ cells per 1 mm^2^ was calculated using the formula:$$N\;\mathrm{in}\;1\;{\mathrm{mm}}^{2}=\frac{\sum N\;\times 1,000,000\;{µm}^{2}}{N\;fields\;of\;view\times S\;of\;one\;field\;({µm}^{2})},$$
where ΣN represents the total number of CD+ cells in all the examined fields.

Then, the obtained morphometric data were converted into scores ranging from 1 to 3, where 1 point represents less than 10 CD+ cells, 2 points represent 10 to 25 CD+ cells, and 3 points represent 25 to 45 CD+ cells for CD3, CD4, CD20, and CD138.

CD68 and CD163 were converted into scores ranging from 1 to 3, where 1 point represents 25 to 55 CD+ cells, 2 points represent 55 to 85 CD+ cells, and 3 points represent 85 to 105 CD+ cells.

Immunohistochemical reactions for interleukins were performed manually using primary antibodies (ThermoFisher, 1:100, Waltham, MA, USA) against IL-1 beta, IL-4, IL-6, and IL-10. Universal antibodies (HiDef Detection™ HRP Polymer system, Cell Marque, Rocklin, California, USA) were used as secondary antibodies. Cell nuclei were counterstained with Mayer’s hematoxylin. The count of immunopositive cells was performed in 10 randomly selected fields of view at a magnification of ×400 (in %).

In real-time polymerase chain reaction (RT-PCR) analysis of the vermiform appendixes of children from different age groups confirmed with COVID-19, SARS-CoV-2 was detected in all samples. In Group I, an increase in ACE2 expression by 1.5 times (8.62 ± 0.85 vs. 5.6 ± 0.42) and an increase in Furin expression by 1.06 times (38.28 ± 1.42 vs. 36.1 ± 1.18) compared to the control group were observed. No correlation with age was found. In Group II, minor changes compared to Group III (control) were detected, which were not significant in this study [15].

Fluorescent in situ hybridization (FISH) was used for the visualization of viral RNA using the RNA scope method according to the manufacturer’s recommendations. Initially, staining was performed using the Multiplex Fluorescent Detection Kit v2 (Sigma-Aldrich, Saint Louis, MO, USA) according to the manufacturer’s protocols. The slides were air-dried at 55 °C overnight and then pre-treated with hydrogen peroxide followed by target retrieval reagent for 3 min in a steam generator. Subsequently, the slides were incubated with protease III (Sigma-Aldrich, Saint Louis, MO, USA) at 40 °C for 15 min. The probe combination for RNA target detection was hybridized at 40 °C for 2 h. Probes in channel C4 were developed using the RNA scope 4-Plex auxiliary kit (Sigma-Aldrich, Saint Louis, MO, USA). For signal amplification, Opal 520 (Sigma-Aldrich, Saint Louis, MO, USA), Opal 570 (Sigma-Aldrich, Saint Louis, MO, USA), and Opal 690 (Sigma-Aldrich, Saint Louis, MO, USA) dyes were sequentially applied. Negative control using the dapB gene probe of the SMY Bacillus subtilis strain and the 3-plex Negative Control reagent (Sigma-Aldrich, Saint Louis, MO, USA) were used to assess background staining. Subsequently, DAPI staining (Sigma-Aldrich, USA) was performed to visualize cell nuclei. The slides were covered with Mount Solid medium (Sigma-Aldrich, Saint Louis, MO, USA), and visualization was carried out using the Leica STELLARIS confocal microscopy system.

Statistical analysis was performed using SPSS 12.0 software (IBM Analytics, Foster City, CA, USA) and rank-based analysis of variance (ANOVA). Student’s t-test was used for data analysis. Differences between samples were considered statistically significant at *p* < 0.05.

## 4. Results

Histological examination of all vermiform appendix samples in children in Group I revealed the morphological pattern of acute appendicitis, including catarrhal (*n* = 1) and destructive forms, such as phlegmonous–ulcerative (*n* = 33) and gangrenous (Table 2).

During the macroscopic examination, the appendixes in children with COVID-19 did not show any differences compared to the control group. Microscopic examination of the samples revealed findings such as congested blood vessels with microthrombi, mild lymphocytic infiltration of the mucous membrane, hyperplasia of lymphoid tissue, and the presence of fecalith.

In the majority of microslides in the appendixes (*n* = 33), phlegmonous and phlegmonous–ulcerative appendicitis was observed during the pathological examination. Macroscopically, the appendixes were enlarged, the serous membrane appeared dull and hyperemic, and the wall was edematous with purulent infiltration spreading to the base and focal necrotic areas in the submucosa. Microscopically, the findings included wall edema, congested blood vessels with microthrombi, crypt dilation, focal stromal hyalinization, diffuse leukocytic infiltration of the mucosa and submucosa primarily composed of neutrophils, and areas of ulceration and erosion (Table 3, Figure 1).

Eight children were found to have gangrenous appendicitis. Macroscopically, the appendix was enlarged, edematous, hyperemic, and presented with ulcerations; the wall and base showed infiltration and areas of necrosis. Microscopically, there was blurred demarcation between the layers, wall edema, congested blood vessels, massive lymphocytic infiltration, areas of coagulative necrosis with erosions (in two microslides, mucosal necrosis), and manifestations. Pus with a hemorrhagic component was visualized in the lumen of the appendixes (Figure 1).

It is worth noting that all microslides, regardless of the appendicitis form, exhibited microthrombi, fibrinoid necrosis of vascular walls, and perivascular lymphocytic infiltration.

During the analysis of appendixes in patients in Group II, all clinical–morphological forms were identified: catarrhal (*n* = 17), phlegmonous (*n* = 26), gangrenous (*n* = 8), and perforated (*n* = 4) (Table 3, Figure 2).

During the morphological examination of appendixes in children in Group III, no signs of pathological processes were detected.

Immunohistochemical analysis revealed the presence of CD-positive cells in the mucosa of appendixes in children in all studied groups, with varying quantities (Table 4 and Table 5). In the immunohistochemical investigation, an increase in the number of CD-positive cells was observed in the mucosa of appendixes and mesoappendix in children in Groups I and II compared to the control group (Group III). Moreover, a predominance of the studied markers was observed in children with COVID-19 (Table 4 and Table 5, Figure 3).

In terms of the distribution area, the proportion of CD-positive immunocompetent cells varied within Group I, with the highest values found for CD3 (T-lymphocyte subpopulation), CD68 (functionally active macrophages), CD163 (M2-macrophages), CD20 (B-lymphocyte subpopulations), and CD138 (plasma cells). However, the labeling for CD4 (T-helper cells, monocytes, macrophages, dendritic cells) in the perivascular space was lower. Additionally, the number of CD68-positive macrophages was significantly increased in children with acute appendicitis without COVID-19 infection.

The marker of T-lymphocytes (CD3) was visualized in immunocompetent cells of the mucosa of appendixes in children in all groups. However, during the novel coronavirus infection, the subpopulation of immunocompetent cells was increased compared to other groups. The children in this group were given a score of 3 on a 4-point rating scale, and the average number of inflammatory cells in this subpopulation was 18.2 ± 0.4 cells per 1 mm^2^ (Table 4, Figure 3).

The labeling of CD4+ T-helper cells was observed in the immunocompetent cells of the mucosa of appendixes in children in all groups. Similarly, during the novel coronavirus infection, the subpopulation of immunocompetent cells was increased compared to other groups. The children in this group were given a score of 2 on a 4-point rating scale, and the average number of inflammatory cells in this subpopulation was 11.2 ± 0.2 cells per 1 mm^2^ (Table 4, Figure 3).

The labeling of CD20+ B-lymphocytes was observed in the immunocompetent cells of the mucosa of appendixes in children in all groups. Likewise, during the novel coronavirus infection, the subpopulation of immunocompetent cells was increased compared to other groups. The children in this group were given a score of 4 on a 4-point rating scale, and the average number of inflammatory cells in this subpopulation was 36.7 ± 2.1 cells per 1 mm^2^ (Table 4, Figure 3).

The labeling of CD138+ plasma cells was observed in the immunocompetent cells of the mucosa of appendixes in children in all groups. Similarly, during the novel coronavirus infection, the subpopulation of immunocompetent cells was increased compared to other groups. The children in this group were given a score of 3 on a 4-point rating scale, and the average number of inflammatory cells in this subpopulation was 19.6 ± 0.4 cells per 1 mm^2^ (Table 4, Figure 3).

The labeling of CD163+ macrophages was observed in the immunocompetent cells of the mucosa of appendixes in children in all groups. Likewise, during the novel coronavirus infection, the subpopulation of immunocompetent cells was increased compared to other groups. The children with COVID-19 were given a score of 3 on a 3-point rating scale (Table 4, Figure 3).

The labeling of CD68+ macrophages was observed in the immunocompetent cells of the mucosa of appendixes in children in all groups. Likewise, during the novel coronavirus infection, the subpopulation of immunocompetent cells was increased compared to other groups. The children with COVID-19 were given a score of 2 on a 3-point rating scale (Table 4, Figure 3).

During immunohistochemical analysis, IL-positive cells were detected in the mucosal and submucosal layers of the appendix in pediatric patients across all investigated groups, primarily in the epithelium and immune cells, with their quantity varying depending on the clinical and morphological forms and the presence of SARS-CoV-2 (Table 5).

Immunohistochemical analysis revealed an increase in the number of IL-1 and IL-6-positive cells in the mucosal layer of the appendixes and mesoappendix of children in Groups I and II compared to the control group (Group III). Meanwhile, a predominance of these markers was observed in children with COVID-19 (Table 5, Figure 4). In cases of novel coronavirus infection, there was a significant increase in the expression of pro-inflammatory cytokines (IL-1, IL-6), which was relatively compensated by similar levels of anti-inflammatory cytokines (IL-4, IL-10) (Table 5).

When assessing the area distribution, the proportion of IL-positive immune cells varied in Group I, with the highest values observed for IL-1 and IL-6. When evaluating microslides based on the clinical and morphological forms of acute appendicitis, the highest distribution of IL-1 and IL-6 was observed in the gangrenous form, followed by the phlegmonous form, compared to the catarrhal form.

Immunohistochemical examination of intact appendixes demonstrated scattered stained inflammatory cells in the tissue, with the proportion of cells expressing pro-inflammatory (IL-1, IL-6) and anti-inflammatory (IL-4, IL-10) cytokines not exceeding 10%. This level can be considered physiological. The balance between pro-inflammatory and anti-inflammatory cytokines was maintained in all age groups (Table 5, Figure 4).

For the immunohistochemical detection of cells expressing the pro-inflammatory cytokine IL-1, antibodies against IL-1β were used as the most commonly used marker of inflammatory response. IL-1 expression was observed in the cytoplasm of epithelial cells and inflammatory cells of the mucosal layer of the appendix, predominantly in mononuclear and polymorphonuclear cells, in all groups. However, during the novel coronavirus infection, the subpopulation of immune cells expressing IL-1 was increased compared to the other groups, especially the control group. The quantity of IL-1-expressing cells sharply increased in all age groups during novel coronavirus infection and was 6.1 times higher than the corresponding control values. When comparing the expression of IL-1 between Group II and Group III, a 2.5-fold increase was observed depending on the clinical and morphological forms of acute appendicitis, with significantly higher levels in the gangrenous and perforated forms (Table 5, Figure 4).

The expression of the pro-inflammatory cytokine IL-6 was analyzed to assess the inflammatory response during immunohistochemical analysis. IL-6 expression was observed in the cytoplasm of epithelial cells and inflammatory cells of the mucosal layer of the appendix, predominantly in mononuclear and polymorphonuclear cells, in all groups. Similar to IL-1, the subpopulation of immune cells expressing IL-6 was increased during novel coronavirus infection compared to the other groups, particularly the control group. The quantity of IL-6-expressing cells significantly increased in all age groups during novel coronavirus infection, exceeding the control values of the corresponding age group by 7.7 times. When comparing the expression of IL-6 between Group II and Group III, a 4.1-fold increase was observed depending on the clinical and morphological forms of acute appendicitis, with significantly higher levels in the gangrenous and perforated forms (Table 5, Figure 4).

The expression of the anti-inflammatory cytokine IL-4 was assessed to analyze the inflammatory response during immunohistochemical analysis. IL-4 expression was observed in the cytoplasm of epithelial cells and inflammatory cells of the mucosal layer of the appendix, predominantly in mononuclear and polymorphonuclear cells, in all groups. As with IL-1 and IL-6, the subpopulation of immune cells expressing IL-4 was increased during novel coronavirus infection compared to the other groups, especially the control group. The quantity of IL-4-expressing cells increased in all age groups during novel coronavirus infection and exceeded the control values of the corresponding age group by 3.8 times. When comparing the expression of IL-4 between Group II and Group III, a 2.7-fold increase was observed depending on the clinical and morphological forms of acute appendicitis, with significantly higher levels in the gangrenous and perforated forms (Table 5, Figure 4).

To analyze the inflammatory reaction in immunohistochemical studies, the expression of the anti-inflammatory cytokine IL-10 was evaluated. In all groups, IL-10 expression was observed in the cytoplasm of epithelial and immunocompetent cells (mononuclear and polymorphonuclear cells) of the appendixes, as well as in the germinal centers of lymphoid follicles in the mucosal and submucosal layers. Meanwhile, during new coronavirus infection, the subpopulation of immunocompetent cells was increased compared to the other groups, especially the control group. The number of cells expressing this cytokine significantly increased in all age groups during new coronavirus infection, surpassing the control values of the same age by a factor of 3.5. When comparing IL-10 expression between Group II and Group III, an increase of 2.3 times was observed depending on the clinical and morphological form of acute appendicitis, with significantly higher levels in cases of gangrenous and perforated appendicitis (Table 5, Figure 4).

Statistically significant differences were observed between the COVID-19 group and the control group (*) compared to the previous age group in the investigated groups (**) with a *p*-value < 0.05.

Thus, immunohistochemical evaluation of pro-inflammatory cytokines (IL-1, IL-6) and anti-inflammatory cytokines (IL-4, IL-10) revealed a shift in the balance toward inflammation. This could be associated with the activation of adaptive mechanisms during SARS-CoV-2 invasion.

In confocal microscopy of the appendixes of pediatric patients with new coronavirus infection, after fluorescence in situ hybridization, a positive signal of SARS-CoV-2 viral RNA was observed in the cytoplasm of the majority of enterocytes in the single-layered columnar epithelium, as well as in individual immunocompetent cells in all examined samples. The signal intensity remained consistent in all microslides in Group I, regardless of age (Table 6).

It is important to note that upon a detailed examination of each age group, specimens were found in which the number of cells with a positive signal significantly deviated from the arithmetic mean for that group, which did not correspond to the normal value. To determine the significance of differences between age groups, the Kruskal–Wallis H-test was used, yielding a value of 5.46 (*p* = 0.065). The differences in the expression of SARS-CoV-2-positive cells in the three examined age cohorts were found to be statistically insignificant.

Microscopic examination revealed that SARS-CoV-2 primarily replicates in the epithelium of intestinal crypts and the mucous membrane of the appendiceal vermiform appendix. Infected cells showed predominantly perinuclear distribution of SARS-CoV-2 RNA (Figure 5).

During confocal microscopy of the vermiform appendages of children in Groups II and III, no presence of SARS-CoV-2 was detected after fluorescent in situ hybridization.

## 5. Discussion

This study describes the morphological features and local immune cell response in acute appendicitis among children of different age groups with confirmed COVID-19 compared to acute appendicitis without COVID-19 infection.

Most microslides exhibited destructive changes in the vermiform appendages. We identified signs indicative of SARS-CoV-2-associated appendicitis, such as microthrombi, fibrinoid necrosis of the vascular wall, and perivascular lymphocytic infiltration, which were observed in all samples in Group I and were not characteristic of the classical morphological picture of acute appendicitis.

Immunohistochemical examination of the vermiform appendages in children with confirmed COVID-19 revealed a predominance of CD3+, CD4+, CD20+, CD138+, CD 163+, and CD68+ cells compared to acute appendicitis without COVID-19 infection and the control group.

The increased number of CD3-positive cells in the mucosa of the vermiform appendage and mesoappendix in Groups I and II indicates immune response activation.

The presence of perivascular lymphocytic inflammation is confirmed by an increase in the number of memory T-cells (CD4+) specific to the viral antigen of SARS-CoV-2. Additionally, the induction of chemokines and cytokines, facilitating the transition of T-lymphocytes from blood to tissues, serves as confirmation [16]. A case described in the literature drew attention to the high correlation between CD3+ and CD4+ immune cells in response to COVID-19 infection [17].

Thus, the significant increase in CD3+ and CD4+ T-cells observed does not exclude the presence of multisystem inflammatory syndrome or its forms in children with COVID-19 across different age groups.

Macrophage activation in tissues leads to hyperinflammation, caused by excessive proliferation of T-lymphocytes [18]. The increase in the number of CD68-positive cells and CD163-positive cells (macrophages) in both groups with acute appendicitis may indicate containment of the infectious agent or an undesirable effect of an inflammatory reaction based on immune system dysregulation.

In response to antigens, CD4+ T-cells produce cytokines that subsequently activate B-lymphocytes (CD20+). In Group I samples, an increase in CD20-positive cells can be observed, indicating the development of humoral immunity at the site of viral genome replication.

The literature describes CD138 as a marker of plasma cells [19], and its increase in the vermiform appendage mucosa of children with COVID-19 indicates elevated antibody production against a new coronavirus infection.

A recent study provided information that the number of B-lymphocytes in the vermiform appendage tissue was higher compared to T-lymphocytes in patients with confirmed COVID-19 [20]. However, our performed in situ hybridization study indicates a predominance of T-cell immune responses in the vermiform appendage tissue in response to SARS-CoV-2 invasion.

The increase in the number of CD138- and CD20-immune cells suggests that children diagnosed with COVID-19 develop antiviral immune responses in the appendix and peri-appendicular tissue. Furthermore, indirect confirmation of our theory is supported by the fact that in Group II, the number of CD20- and CD138-positive cells remains lower than in Group I. In acute appendicitis, one of the leading factors in inflammation development is the bacterial intestinal microflora, which is activated in response to the blockage of the appendix lumen by enlarged lymphoid tissue, triggering a cellular immune response.

One of the objectives of our study was to molecularly and biologically evaluate the levels of pro-inflammatory cytokines (IL-1, IL-6) and anti-inflammatory cytokines (IL-4, IL-10) in the appendixes of children with COVID-19, considering the lack of data on this in the global literature.

IL-1 and IL-6 are known to be key cytokines in innate immunity, and they induce the development of a “cytokine storm”, thereby increasing the migration of immune cells to the site of inflammation [21].

In a meta-analysis by Sujan K. Dhar et al., the significance of elevated levels of IL-1 in the blood of patients with novel coronavirus infection could not be demonstrated [22]. However, in our study, we observed an increase in IL-1 concentration in the appendixes of children with confirmed COVID-19, which was 1.3 times higher compared to acute appendicitis and 3.5 times higher compared to appendixes during non-pandemic periods.

Grifoni, E. and others have shown that IL-6 is detected in high levels in the blood of patients with COVID-19 [23]. We also observed an increase in IL-6 in the appendixes of children with COVID-19, which may lead to a decrease in viral lytic activity [24].

Our study particularly focused on the concentration of anti-inflammatory cytokines (IL-4, IL-10) in the appendixes of children with COVID-19.

Previous studies have described that an increase in IL-10 concentration in the blood indicates the peak of the “cytokine storm”, limiting the inflammatory process [25]. Similarly, we observed an increase in IL-10 in the appendixes of children with confirmed COVID-19 compared to acute appendicitis and appendixes during non-pandemic periods.

IL-4 is known to inhibit the production of pro-inflammatory cytokines, such as IL-1 and IL-6. In a study by Vaz de Paula CB et al., it was shown that the level of IL-4 is significantly elevated in the blood and lung tissues of patients with COVID-19 [26]. According to our findings, IL-4 was also elevated in the appendixes of children with COVID-19 compared to acute appendicitis and appendixes during non-pandemic periods.

According to the literature, the ratio of IL-6/IL-10 is a prognostic indicator of a severe course of coronavirus infection [22,27].

Thus, in children with COVID-19 and appendicitis, an increase in pro-inflammatory cytokines IL-1 and IL-6, as well as anti-inflammatory cytokines IL-4 and IL-10, is observed. At the same time, there is a predominance of the IL-6/IL-10 ratio, which likely indicates the severity of the infection.

In the samples from Group I tissues, the presence of SARS-CoV-2 was detected by PCR-RV, indicating the tropism of the virus for the appendix.

There is a possibility that with the penetration of the virus into the cell, there is specific viral damage to the tissues. As a result, SARS-CoV-2 nucleic acids gain access to the synthesis mechanisms in the cytoplasm and nucleus of epithelial cells. According to our data, the development of SARS-CoV-2-associated appendicitis in children cannot be excluded.

Thus, SARS-CoV-2 exerts a direct damaging effect on the organ, invading the cells through ACE-2 and Furin proteins, as confirmed by the high expression levels of ACE-2 and Furin [28].

In our study, children with confirmed COVID-19 diagnosis showed an increase in the expression of the gene-encoding components of the ACE2 receptor complex, indicating an increased likelihood of SARS-CoV-2 penetration and aggressive impact on the appendix tissue. It is known that ACE2 also plays an important role in the regulation of blood pressure and hemodynamics [29,30].

A slight increase in the expression of Furin in the appendix tissue in the samples in Group I was described in the literature as a predisposing factor for SARS-CoV-2 cell penetration [31].

In all control samples analyzed by confocal microscopy, we did not detect a positive signal, indicating the successful adaptation of the FISH method in this study, which excludes the presence of false-positive results.

In our study, a record number of cells with positive viral RNA signals were detected in the appendix tissue of children in Group I. No significant differences in the number of infected cells in the appendix were found when comparing different age groups. Initially, it was hypothesized that ACE2 expression increases with age, making the adult population more vulnerable to SARS-CoV-2. However, it is now known that ACE2 expression is higher in the intestines of children [32].

Thus, the detection of SARS-CoV-2 RNA in all samples of appendix tissue from children who have had COVID-19 suggests tropism of the SARS-CoV-2 virus to the epithelium of the appendix.

Summarizing our findings, it can be concluded that there is tropism of the COVID-19 virus to the epithelium of the appendix, as evidenced by the detection of SARS-CoV-2 RNA using PCR-RV and FISH methods, as well as the presence of local inflammatory reactions. However, further research in this direction is needed.

## 6. Conclusions

In the appendix of children with COVID-19, a morphological picture of destructive forms of acute appendicitis (phlegmonous–ulcerative and gangrenous) is observed, as well as congestion of blood vessels and microthrombi. SARS-CoV-2 invasion, an increased expression of ACE2 and Furin, and an increase in the number of CD3+, CD4+, CD20+, CD68+, CD163+, and CD138+ cells were detected, indicating the activation of both cellular and humoral local immunity. In addition, there was an increase in pro-inflammatory cytokines IL-1 and IL-6, as well as anti-inflammatory cytokines IL-4 and IL-10, with a predominance of the IL-6/IL-10 ratio. The aforementioned pathological and immunohistochemical changes were more pronounced in the group of children aged 6-12 years (childhood).

## Figures and Tables

**Figure 1 biomedicines-12-00312-f001:**
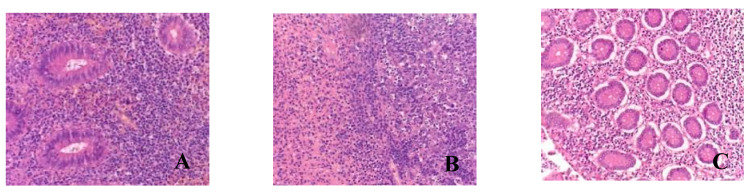
Morphological picture of acute appendicitis in children with COVID-19: (**A**)—catarrhal; (**B**)—phlegmonous–ulcerative; (**C**)—gangrenous. Staining: hematoxylin and eosin; magnification ×200.

**Figure 2 biomedicines-12-00312-f002:**
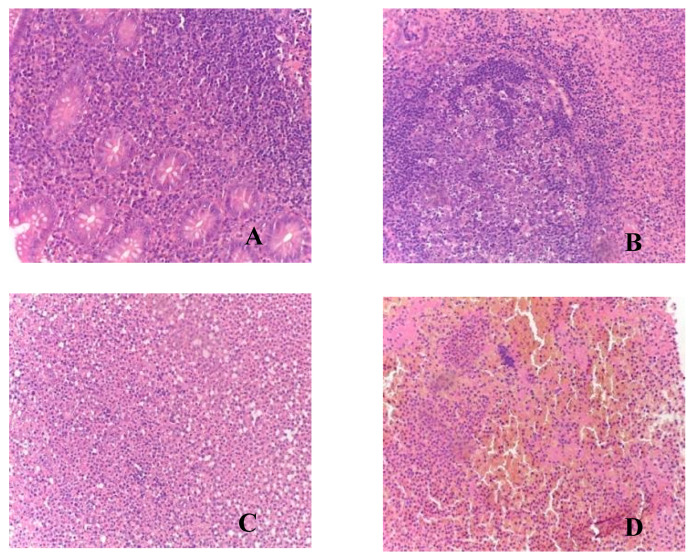
Morphological picture of acute appendicitis in children in Group II: (**A**)—catarrhal; (**B**)—phlegmonous; (**C**)—gangrenous; (**D**)—perforated. Staining: hematoxylin and eosin; magnification ×200.

**Figure 3 biomedicines-12-00312-f003:**
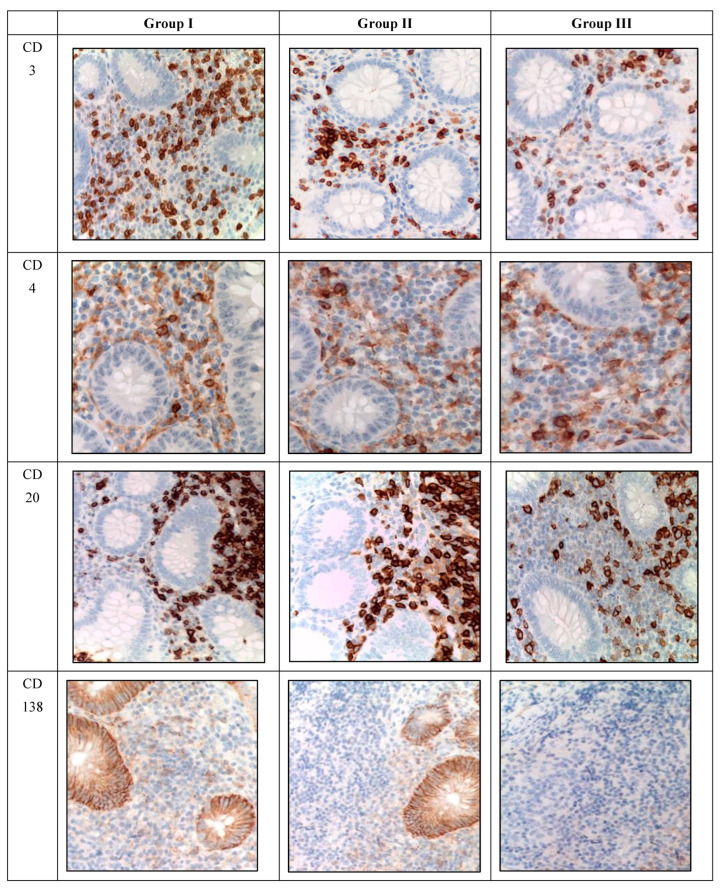
Appendixes in children in the studied groups. Immunohistochemical reactions with antibodies against CD3, CD4, CD20, CD138, CD68, and CD163 counterstained with hematoxylin; magnification ×400. Expression in immunocompetent cells of the mucosa.

**Figure 4 biomedicines-12-00312-f004:**
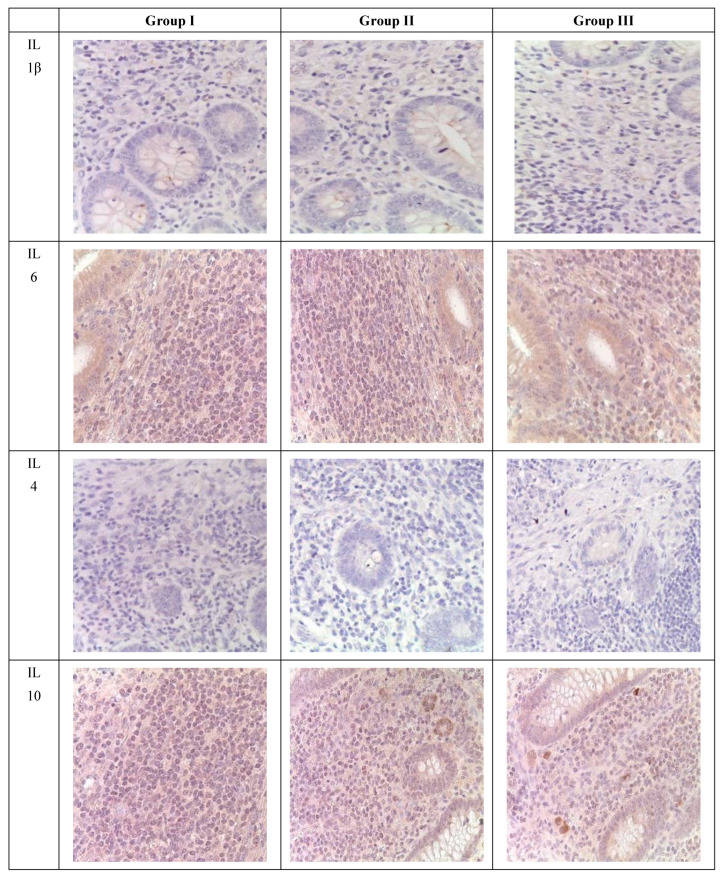
Appendixes in children from the investigated groups. Immunohistochemical reactions with antibodies against IL-1β, IL-6, IL-4, IL-10, counterstained with hematoxylin; magnification ×400. Expression in immunocompetent cells of the mucosal layer.

**Figure 5 biomedicines-12-00312-f005:**
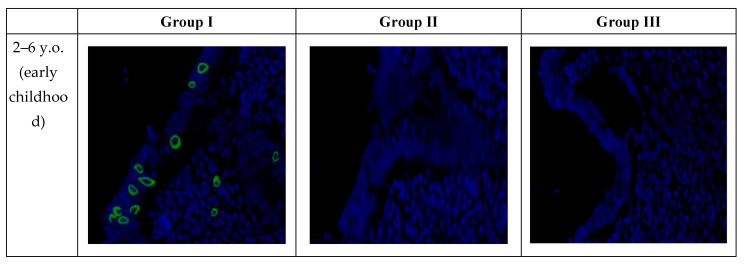
SARS-CoV-2 RNA in the appendixes of children detected by fluorescence in situ hybridization; magnification ×200.

**Table 1 biomedicines-12-00312-t001:** Age groups according to child age classification [14].

Age	Subgroup	*n*
2–6 y.o. (early childhood)	I—COVID-19	9
	II—appendicitis	15
	III—control	8
6–12 y.o. (childhood)	I—COVID-19	13
	II—appendicitis	19
	III—control	13
12–18 y.o. (teens)	I—COVID-19	20
	II—appendicitis	21
	III—control	17

**Table 2 biomedicines-12-00312-t002:** Clinical–morphological forms of acute appendicitis in children in Group I.

Group/Form	*n*	Catarrhal	Phlegmonous–Ulcerative	Gangrenous
2–6 y.o. (early childhood)	9	1	7	1
6–12 y.o. (childhood)	13	0	10	3
12–18 y.o. (teens)	20	0	16	4
Total:		1	33	8

**Table 3 biomedicines-12-00312-t003:** Clinical–morphological forms of acute appendicitis in children in Group II.

Group/Form	*n*	Catarrhal	Phlegmonous	Gangrenous	Perforated
2–6 y.o. (early childhood)	15	9	5	1	0
6–12 y.o. (childhood)	19	5	13	1	0
12–18 y.o. (teens)	21	2	13	4	2
Total:		16	31	6	2

Group/Form *n*: catarrhal, phlegmonous, gangrenous, and perforated.

**Table 4 biomedicines-12-00312-t004:** Number of CD-positive cells per 1 mm^2^, scored.

Group	*n*	CD3	CD4	CD20	CD138	CD68	CD163
COVID-19	11	3	2	3	3	2	3
Acute appendicitis	26	2	1	2	1	1	2
Control	51	1	1	1	1	1	1

**Table 5 biomedicines-12-00312-t005:** Expression of pro-inflammatory and anti-inflammatory cytokines in patients with novel coronavirus infection (COVID-19) and during physiological spermatogenesis (control) per 1 mm^2^, %.

Group	*n*	IL-1	IL-6	IL-4	IL-10
COVID-19					
catarrhal	1	11.6	12.6	10.8	21.3
phlegmonous–ulcerative	33	22.7	31.1	16.2	34.5
gangrenous	8	48.6	58.9	25.3	46.9
Acute appendicitis					
catarrhal	16	8.5	10.2	9.3	18.9
phlegmonous	31	18.3	27.6	12.4	29.7
gangrenous	6	34.7	47.8	24.8	41.5
perforative	2	43.8	62.4	36.1	52.4
Control	38	8.3	9.5	8.6	15.4

**Table 6 biomedicines-12-00312-t006:** Average distribution of cells with detected SARS-CoV-2 RNA across age groups, in %.

Age	Subgroup	RNA SARS-CoV-2
2–6 y.o. (early childhood)	I—COVID-19	34.26 ± 10.7
6–12 y.o. (childhood)	I—COVID-19	33.78 ± 8.7
12–18 y.o. (teens)	I—COVID-19	34.68 ± 9.3

## Data Availability

The study did not generate publicly available archival data.

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
