# Peer review of "Features of Appendix and the Characteristics of Appendicitis Development in Children with COVID-19"

_biomedicines, 2024, doi:10.3390/biomedicines12020312_

Round 1

Reviewer 1 Report

Comments and Suggestions for Authors

Reviewer’s comments to Author:

1.  The title: Features of Appendix Development in Children with COVID-19. Is it possible to consider revising it to "Features of Appendix And the Characteristics of Appendicitis Development in Children with COVID-19", which would be more appropriate to study the content of the manuscript?

2.    The number of cases analyzed by each group in this research manuscript is relatively small. It would be a reference and valuable study if clinical cases could be collected on a larger scale or in multi-centers for further analysis.

3.    In the discussion sections, the author is asked to briefly describe why patients infected with COVID-19 in "Clinical-morphological forms of acute appendicitis in children of Group I" can cause varying degrees of catarrhal, phlegmonous-ulcerative and gangrenous appendicitis based on this research conclusions and pathophysiological perspectives and pathological mechanism?

4.      Please briefly describe the strengths, weaknesses, and limitations of your approach to conducting this study.

Author Response

  1. We agree with your proposal, we will change the name
  2. We agree with your comment, but unfortunately it was not possible to make a larger sample.
  3. In the discussion, we talk about the parameters by which patients in groups 1 and 2 differ. Apparently these differences lead to the highest incidence of phlegmonous-ulcerative appendicitis in children with COVID-19. This manuscript has a morphological focus, and these are the criteria we considered.
  4. Strengths: lack of data in the scientific literature on forms of appendicitis in children with COVID-19. Weaknesses: small sample of children.

Reviewer 2 Report

Comments and Suggestions for Authors

Dear Authors,

I found your paper interesting. It is a  retrospective insight in the immunopathology of appendix involvement by SASRS CoV2 virus. This is a new matter, poorly described in literature, useful for surgeons to be aware of the severity of appendicites in COVID+ children. The article is well written, clear methods and results, good discussion and conclusions supported by data. I have minor suggestions: please add the stadard deviations when reporting age; minor language polishing by a mother tongue could improve the article readability; I am not sure to have figured out figure 5: I would not expect fluorescence in patients of group II and III, why then all the 9 subfigures are fluorecence positive?

Comments on the Quality of English Language

Minor english polishing by a mother tongue would help improve the readability.

Author Response

1. please add the standard deviations when reporting age. 

Standard deviations added

2. I am not sure to have figured out figure 5: I would not expect fluorescence in patients of group II and III, why then all the 9 subfigures are fluorecence positive? 

The picture has been changed, for technical reasons the pictures were mixed up when sending